# The Regulatory Axis of PD-L1 Isoform 2/TNF/T Cell Proliferation Is Required for the Canonical Immune-Suppressive Effects of PD-L1 Isoform 1 in Liver Cancer

**DOI:** 10.3390/ijms24076314

**Published:** 2023-03-28

**Authors:** Xixi Zheng, Xingdong Chen, Weicheng Wu

**Affiliations:** 1State Key Laboratory of Genetic Engineering, Human Phenome Institute, School of Life Sciences, Fudan University, Shanghai 200438, China; 2Taizhou Institute of Health Sciences, Fudan University, Taizhou 225316, China; 3Rugao Joint Research Institute of Longevity and Aging, Fudan University, Rugao 226599, China

**Keywords:** liver cancer, programmed cell death protein-ligand 1, immunology suppression, isoforms

## Abstract

Despite the well-studied effects of the full-length membrane-locating isoform Iso1 of Programmed Cell Death Protein-Ligand 1 (PD-L1) on immunosuppression, little is known about another membrane-locating isoform, Iso2. While expressional and survival analysis of liver cancer patients indicated that Iso2 plays a tumor-suppressive role, our results also indicated that the tumor-promoting and immune-suppressive effects of Iso1 depended on the positive expression of Iso2. Through mediation analysis, we discovered several downstream genes or pathways of Iso2 and investigated their effects on the Iso1-regulating survival. Among all potential downstream immune factors, Iso2 was inclined to activate the proliferation of T cells by regulating chemokine activity and increasing CD3 levels by promoting TNF expression. Similar results were confirmed in the Mongolian liver cancer cohort, and the Iso2/TNF/T-cell axis was verified in several other cancers in the TCGA cohort. Finally, we demonstrated the promoting effects of Iso2 in terms of producing TNF and increasing T cells both in vitro and in vivo. Our findings illustrate that PD-L1 Iso2 can increase the number of T cells in the tumor microenvironment by elevating TNF levels, which is a necessary part of the tumor-suppressive effects of Iso1 in liver cancer.

## 1. Introduction

Programmed cell death ligand-1 (PD-L1), also known as cluster of differentiation 274 (CD274), is a co-inhibitory factor of the immune response. Its canonical full-length isoform (Iso1), mainly localizing on the cell membrane, can reduce the proliferation of T cells and inducing apoptosis by interacting with the T-cell membrane-locating inhibitory checkpoint molecule PD-1 [1]. Several other isoforms of PD-L1 exert different biological roles. For instance, nuclear PD-L1 (nPD-L1) interacts with the cohesin complex and regulates sister chromatid cohesion [2], while exosome-localized PD-L1 (exo-PD-L1) influences intercellular communication, as well as the composition and dynamics of the extracellular environment [3]. Some of these isoforms, such as exo-PD-L1 and soluble PD-L1 (sPD-L1), confer similar immunosuppressive effects to Iso1 due to their extracellular location and potential interaction with PD-1 [4]. Intriguingly, an IGV-deficient membrane-locating isoform (Iso2), which was generated by the excision of exon 2 encoding IgV during pre-mRNA editing [5], was also observed in an in vitro study, suppressing the immune response by regulating the apoptosis and proliferation of T cells [6]. Since the IgV domain is critical for the interaction of PD-L1 and PD-1 [7], the lack of this domain in Iso2 does not support its binding with PD-1 and its consequential suppressive effects on T cells. Therefore, whether Iso2 is involved in immunosuppression, how Iso2 works, and whether Iso2 interacts with Iso1 during this process are all topics in need of further clarification.

In the original report on Iso2, a tumor-promoting effect of Iso2, similar to that displayed by Iso1 and sPD-L1, was observed in colorectal cancer patients [6]. The tumor-promoting roles of Iso1 have been widely verified, as demonstrated by the great success of anti-PD-L1 and anti-PD-1 therapies in various cancers [7]. However, there are still conflicting reports on the tumor-suppressive effects of PD-L1 in various cases. Consider liver cancer, for example: despite the positive association between PD-L1 expression and tumor aggressiveness [8], it is reported that the patients with high expression of PD-L1 and a high tumor infiltrating lymphocytes (TILs) presence have a better prognosis [9]. Considering the antibodies used to determine the expression of PD-L1 might also recognize Iso2, we have reason to speculate that Iso2 might have distinct or even opposing effects on tumor progression.

Unlike other malignancies, there are currently no acknowledged biomarkers with a stable, predictable effect on resistance or response to immunotherapy in liver cancer patients. The role of PD-L1 expression in liver cancer remains controversial. In the KEYNOTE-224 trial, the expression of this ligand was associated with improved object response rate and progression-free survival (PFS) in patients responding to therapy [10]. Additionally, in the IMbrave 150 trial, patients with high levels of PD-L1 mRNA also showed PFS [11]. However, no difference was observed in the PFS and overall survival (OS) of patients with high baseline PD-L1 expression ≥ 1, at least according to the CheckMate 459 trial [12]. Taken together, the predicted value of total PD-L1 expression is limited in liver cancer; hence the need for further studies investigating the distinct expression pattern and biological function of different isoforms. 

Therefore, to investigate the roles of Iso2 and its potential synergistic or antagonistic effects in combination with Iso1, we chose the liver as the object with the highest Iso2 levels among normal tissues and then analyzed the expression patterns, survival effects and related functions of Iso1 and Iso2 in the TCGA-LIHC cohort and a Mongolian liver cancer cohort. Our data revealed that Iso2 elevated TNF levels, thus regulating a series of T-cell-associated functions, particularly the proliferation of T cells. Moreover, these Iso2-downstream T-cell-associated pathways were critical for the canonical effects of Iso1 on inhibiting immunity and promoting tumor progression. Finally, the regulatory axis of Iso2/TNF/T cell proliferation was demonstrated by in vitro and in vivo experiments and was observed in other cancers. In summary, in this study, we revealed the function of Iso2 and preliminarily elucidated the interplay between two PD-L1 isoforms in cancers.

## 2. Results

### 2.1. Different Expression Trends of Two Membrane Localization PD-L1 Isoforms in Liver Cancer

Although the expression and function of Iso2 in colorectal cancer have been reported, its expression patterns in other cancers have yet to be clarified. To elucidate the expression pattern of PD-L1 Iso2, we performed a pan-cancer analysis on the transcriptional isoform percentage of Iso1 and Iso2 in the TCGA and GTEX datasets, and found that Iso2 could be detected in both normal and tumor tissues, with average expression only second to Iso1 (Figure 1A). Among all tissues, normal liver tissue conferred the highest percentage of Iso2. Thus, we chose the liver and its related cancer as the research objects. In the normal/para-cancer/tumor tissues, the average proportion of Iso1 was 39.37%, 65.42% and 56.20%, while that of Iso2 was 38.96%, 19.16% and 17.29%, respectively (Figure 1B). Statistical analysis revealed significant proportional decreases in Iso2 in tumor tissues (tumor vs. normal *p* = 9 × 10^−15^; tumor vs. para-cancer *p* = 0.14) (Figure 1C), as well as significant downregulation of its relative mRNA levels (tumor vs. normal *p* = 9.2 × 10^−25^; tumor vs. para-cancer *p* = 0.031) (Figure 1D). Similar downregulation of Iso2 from para-cancer to tumor tissues was observed in the re-calculated isoform expression data from a Mongolian cohort (GSE144269 [13], Figure 1E).

We also performed fluorescence in situ hybridization (FISH) tests on a tissue microarray (TMA) consisting of 180 pairs of para-cancer and tumor tissues from patients with liver cancer and detected a significant decrease in Iso2 in tumor tissues (Figure 1F,G). Similar to the RNA-seq results from the public cohorts, the FISH-staining pattern of Iso2 was also highly heterogeneous among patients. Strong Iso2 staining was only observed in a few patients and in most tumor tissues, Iso2 was undetectable. Although the expression trends of Iso1 were inconsistent among two public datasets and the FISH results, Iso1 and Iso2 conferred different expression patterns, suggesting their different roles in tumor progression (Figure 1F,G). Meanwhile, FISH data also showed the co-expression of Iso2 and Iso1 in most para-cancer parenchyma cells and tumor cells (Figure 1H and Appendix A), suggesting that there might be functional interplays between Iso1 and Iso2. Despite the dominated location in parenchyma cells and tumor cells, co-expression of these two isoforms could also be observed in some tumor-associated cells with nontumorous morphology. Meanwhile, an immune analysis of TCGA data using different tools revealed that level changes of Iso1 and Iso2 in tumor tissues were significantly associated with the numbers of macrophages and M2-type macrophages (Appendix A), suggesting Iso1 and Iso2 might also function in macrophages.

### 2.2. The Influence of PD-L1 Iso2 on the Prognosis Effects of Iso1 

Next, we performed survival analysis on Iso1 and Iso2 to determine their clinical roles. The Uni-cox data showed that higher levels of Iso2 were significantly associated with better prognosis in Overall Survival (OS) (HR > 0 and *p* = 0.00454), Disease-Specific Survival (DSS) (HR > 0 and *p* = 0.00543) and Progression-Free Interval (PFI) (HR > 0 and *p* = 0.0474), while Iso1 conferred opposite effects, especially in PFI (HR < 0 and *p* < 0.05) (Figure 2A and Appendix A). A similar result was observed in Kaplan–Meier analysis (Figure 2B). Meanwhile, similar hazard ratios of para-tumor Iso2 on PFI were also detected, despite its nonsignificance (Figure 2A). 

Based on the cut-off point in each Kaplan–Meier analysis, we divided all patients into Iso1^high^ Iso2^high^, Iso1^high^ Iso2^low^, Iso1^low^ Iso2^high^ and Iso1^low^ Iso2^low^ groups and observed the significant prognostic difference in PFI among all these groups (*p* = 0.016) (Figure 2C). Intriguingly, patients in the Iso1^low^Iso2^high^ group had the longest median survival time, while those in Iso1^high^Iso2^high^ group had the shortest. A significant prognostic difference was only observed between these two Iso2^high^ groups (*p* = 0.0065) instead of the other two Iso2^low^ groups (*p* = 0.57) (Figure 2C), indicating that the tumor-promoting effects of Iso1 depended on the upregulation of Iso2. 

Considering the negative expression of Iso2 in most liver tumor tissues (Figure 1), we also analyzed the expression distribution of Iso1 and Iso2 and identified zero as a better expression cut-off point for Iso2 to group the patients with liver cancer in the TCGA cohort (Figure 2D). Patients were divided into Iso2 positive and negative groups, and the effects of Iso1 on PFS were analyzed in these two groups. Similar to the result in the Iso2^high/low^ groups, the significant effect of Iso1 on survival disappeared when Iso2 became negative (Figure 2E). We observed a similar result in the Mongolian cohort, in which significant effects of upregulated Iso1 on poor prognosis (*p* = 0.023) were only observed in Iso2-positive patients (*p* = 0.011), while the effect was reversed in Iso2-negative patients (*p* = 0.006). Our data also revealed that negative expression of Iso1 was critical to the tumor-suppressive effects of Iso2 (*p* = 0.001) (Figure 2F). Taken together, we observed opposite roles and potential functional interplays between Iso1 and Iso2 in the prognosis of liver cancer patients.

### 2.3. Associated Functions of PD-L1 Iso2, Including Immune Regulation

To understand the mechanism underlying the opposite effects and the interplays between two isoforms, it is necessary to determine the biological functions of Iso2. Thus, we performed a differential analysis between Iso2-positive and -negative patients, screening out 2199 upregulated genes and 3288 downregulated genes in Iso2-positive patients (Appendix A). The functional enrichment of these genes showed that positive expression of Iso2 was closely associated with protein functions (protein targeting, proteasome-mediated ubiquitination protein catabolism and regulation of protein catabolism), RNA metabolism (RNA catabolism pathway, RNA splicing, purine nucleoside triphosphate metabolism), RNA splicing (spliceosome assembly and U6 SNRNA binding) and several immune pathways, such as changes in T cells’ proliferation or functions, activation of an innate immune response, adaptive immune activation of a lymphocyte-mediated immune response, etc. Consistently, the gene set variation analysis (GSVA) and gene set enrichment analysis (GSEA) also revealed the promoted effects of Iso2 on immune responses, including T cell proliferation, activation and differentiation, T cell receptor binding, leukocyte proliferation and T cell-mediated cytotoxicity (Appendix A). Moreover, in the integrated network analysis of the intersection of 365 Iso2-associated genes and 77 pathways, most hub functions were immune related, and the pathways in the major hub were all associated with leukocyte cell proliferation and activation (Figure 3A and Appendix A). 

### 2.4. Similar and Different Functions between Two mPD-L1 Isoforms 

We also analyzed the differential genes and pathways between Iso1^pos^ and Iso1^Neg^ patients and explored the similarities and differences in the related functions between Iso1 and Iso2. We observed several canonical PD-L1-associated genes significantly associated with Iso1 level changes, such as nuclear factor-κB (NF-κB, *NFKB1*), and several subunits (*AP1S1*, *AP1B1*, *AP1G2*, and *AP1M1*) of activator protein 1 (AP1) assembly protein complex (Appendix A) that were reported to modulate PD-L1 up-regulation via direct transcriptional regulation [14,15]. Iso1 was also correlated with some canonical PD-L1-associated pathways, such as Rho protein signal transduction pathway and phospholipase C-activating G protein-coupled receptor signaling pathway [16]. Enrichment data of differentially expressed genes showed that unlike Iso2, Iso1 was primarily associated with signal transduction (Ras protein signaling and phospholipase C-activated G protein-coupled receptor signaling pathway), calcium regulation (intracellular calcium homeostasis, regulation of cytosolic calcium concentration) and antimicrobial response (Figure 3B,C and Appendix A). Additionally, contrary to the roles of Iso1, GSVA results indicated that Iso2 was negatively correlated with metabolism, protein modification, signal transduction and ion transport, and positively with ubiquitination, ribosome function, spliceosome, transcriptional regulation, snRNA function and apoptosis (Figure 3D,E and Appendix A). Although these two isoforms also conferred opposite correlations with some immune-related pathways, they seemed to play similar roles in the negative regulation of microglia activation, the regulation of B cell proliferation and the regulation of T cell killing (Figure 3D,E). Further GSEA confirmed the opposite roles of Iso2 and Iso1 in ribosome regulation, spliceosome assembly, U6 SNRNA binding and chromosome silencing, as well as their similar roles in T cell receptor complexes, immunoglobulin compounds, immunoglobulin receptors, the immune globulin complex cycle, antigen binding, the regulation of T cell proliferation and apoptosis, and the regulation of alpha–beta T cell differentiation and regulation (Figure 3F,G and Appendix A). These data indicated that despite the opposite roles in diverse biological processes, two PD-L1 isoforms might exert mutual or synergistic effects in immune regulation.

### 2.5. Key Immune Pathways Critical to the Functional Changes of Iso1 Resulting from the Expressional Alternations of Iso2

Considering the potential coordination between two isoforms in immune regulation (Figure 3), and the dependency of the tumor-promoting effect of Iso1 on the positive expression of Iso2 (Figure 2), we speculated that the expressional alternation of Iso2 might change the immune functions of Iso1. Thus, we analyzed the Iso1-correlated genes and GSVA/GSEA pathways in Iso2^Pos^ and Iso2^Neg^ patients, respectively. We obtained 6208 genes, 3993 GSVA pathways and 674 GSEA pathways significantly associated with Iso1 in the Iso2^Pos^ group, and 1724 genes, 1326 GSVA pathways and 509 GSEA pathways in the Iso2^Neg^ group (Figure 4A–C and Appendix A).

Among these candidates, 653 GSVA pathways and 207 GSEA pathways and 353 genes were inversely correlated with Iso1 and Iso2, and after the intersection, 31 genes and 79 enriched pathways were obtained (Figure 4D). Almost all 79 pathways were related to the classical immune regulatory ability of PD-L1, including the secretion of cytokines, immune receptor signal antigen receptors, the immune response (CD4^+^ alpha beta T cell activation, regulating lymphocyte proliferation and NK cell activation, lymphocyte apoptosis, lymphocyte stimulus adjustment), immune damage and immune differentiation (differentiation of T cells, B cells) (Figure 4E and Appendix A). All these genes and pathways conferred opposite correlations with Iso1 between Iso2^Pos^ and Iso2^Neg^ groups, indicating that the immunoregulatory effects of Iso1 could be reversed by Iso2. Since Iso1 is known to promote tumor progression via its canonical immunosuppressive ability, our data partially explain why Iso1 played tumor-promoting roles only in the Iso2-positive group in the survival analysis.

### 2.6. The Downstream Effects of Iso2 on the Proliferation of T Cells Mediated by TNF 

Though we determined that Iso2 affected the immune functions of Iso1, it is still unclear how Iso2 was involved in this process. Additionally, two questions had yet to be answered: what the key immune-related downstream factors of Iso2 were, and whether these downstream factors affect the role of Iso1 in immunosuppression and patient survival. We performed mediation analysis of all 365 genes and 77 pathways that closely related to Iso2 as mediators or effectors (Figure 2A and Figure 5A), elucidating the potential causal relations between these factors and Iso2. Of all the mediation pathways, significant total effects were detected in 63.16% (123,410/195,364), and only 22.18% (43,339/195,364) significant average causal mediation effects (ACMEs) could be observed (Figure 5A). Meanwhile, nearly all candidate pathways had the opportunity to be significant downstream outcome targets (77/77) or key mediators (76/77) of Iso2, and most genes could be effectors (213/365) or mediators (205/365) as well (Appendix A).

To clarify the importance and the universality of these downstream factors, we calculated their frequency detected as mediators or targets in the mediations with significant ACMEs. The immune-related mediator pathway with the top frequency was cytokine activity, and the top target pathway was the positive regulation of active T cell proliferation, indicating that Iso2 was capable of promoting T cell proliferation by changing cytokines activity (Figure 5B,C). Meanwhile, the immune-related mediator gene with the highest frequency was *TNF* (Top9) and the target gene was *CD3D* (Top3). Since CD3 is the pan-marker of T cells, the mediation results of pathways and genes indicated that Iso2 increased the number of CD3^+^ T cells by promoting the proliferation of T cells, which is mediated by the levels and the activity of *TNF* (Figure 5D). We also verified the significant regulation axis of Iso2-TNF-T cell proliferation in both TCGA-LIHC and Mongolia liver cancer cohorts (Figure 5E,F). Moreover, this axis with significant ACME and total effects could be observed in other cancer types in TCGA data, including BRCA, BLCA, ESCA, HNSC, LGG, LUAD, SARC, SKCM, STAD, TGCT and THCA (Figure 5G), indicating that the proliferation of T cells is a universal immune-related downstream function of Iso2.

### 2.7. Roles of the Key Downstream Immune Target of Iso2 in Regulating the Prognosis Effects of Iso1

To confirm that the downstream immune pathway of Iso2, especially the T cell proliferation mentioned above, was critical to the functions of Iso1, we further performed causal mediation analysis to investigate the prognostic effects of Iso1. We selected the intersected immune pathways as mediator candidates, which were downstream of the Iso2 (n = 77, Figure 6A–C) used for mediation analysis (Figure 6A). Our results revealed that despite the lack of direct effects of Iso1 on the survival of patients with liver cancer (Figure 2 and Figure 6), most downstream immune pathways of Iso2, including T cell proliferation, T cell activation, T cell cytotoxicity, etc., significantly mediate the prognostic effects of Iso1 (Figure 6B and Appendix A). Similar effects of T cell proliferation were also observed in the Mongolia liver cancer cohort (Figure 6C). However, we failed to detect its significant mediating effects on Iso1-associated survival in other cancer types (Figure 6D), suggesting that the functional regulatory factors of Iso1 might vary between different cancer types, and the observed interplays between two PD-L1 isoforms in our study might be specific in liver cancer. 

### 2.8. Effects of Iso2 Verified by In Vitro and In Vivo Experiments

Besides T-cell-associated functions, *TNF* itself could be directly or indirectly regulated by Iso2 in the mediation analysis (Figure 5 and Appendix A). Meanwhile, *TNF* levels were only changed by Iso2 instead of Iso1 (Figure 4A), supporting the assumption that *TNF* is the specific downstream factor of Iso2. To verify the regulation of Iso2 on *TNF* and T cell numbers, we overexpressed Iso2 in two liver cancer cell lines (Huh7 and LM3) and detected elevated TNF-α by elisa (Figure 7A). To verify the effects of Iso2 on T cell proliferation, PBMC-derived T cells were co-cultured with Iso2-transfected tumor cells or treated with the culture medium of tumor cells. The results revealed that Iso2-overexpressed tumor cells elevated the total Ki-67 expression and the Ki-67-high proportion of CD3^+^ T cells (Figure 7B), indicating that tumor cell-derived Iso2 promotes T cell proliferation. Meanwhile, the hydrodynamic injection mouse model demonstrated overexpressed Iso2 in some hepatoparenchymal cells, as well as upregulated TNF-α levels in these cells and surrounding niches, leading to stronger inflammation, more inflammatory foci and more recruited immune cells in niches than in the controls (Figure 7C,D). We also detected positive correlation between Iso2 levels and TNF-α levels in human HCC tumor tissues and determined increased numbers of CD3^+^ T cells in Iso2^pos^ tissues, regardless of the expression of Iso1 (Figure 7E,F).

The mediation analysis confirmed that *LAG3* is also one of the top 10 target genes of Iso2 (Figure 8A) and *LAG3* is one of the markers of exhausted T cells [17]. Considering that PD-L1 Iso1 is capable of mediating the exhaustion of T cells, we speculated that the regulation of Iso2 on *LAG3* depends on the expression of Iso1. The mediation analysis confirmed that this regulation was significantly mediated by Iso2 in all patients in the TCGA-LIHC cohort (Figure 8A). Both significant direct and indirect effects from Iso2 to *LAG3* were observed in Iso1^Pos^ patients instead of Iso1^Neg^ patients (Figure 8B,C). Additionally, further immunostaining on human tumor tissues confirmed that more CD8^+^LAG3^+^ exhausted T cells were detected in Iso1^Pos^Iso2^Pos^ niches than Iso1^Pos^Iso2^Neg^ niches (Figure 8C).

Taken together, our results confirm that Iso2 can directly upregulate TNF synthesis in tumor cells, thus increasing T cell numbers in the tumor microenvironment of liver cancer. In the absent of Iso1, increased T cells exert their immune killing effects to eliminate tumor cells, while the positive expression of Iso1 converts most of the T cells to exhausted ones, promoting tumor progression to the contrary. Our data also indicated that Iso2 and its downstream effects on TNF-α and T cells are required as part of the immunosuppressive effects of Iso1, and in Iso2^Neg^ tumors, Iso1 somehow plays tumor suppressive roles (Figure 8F). 

## 3. Discussion

As an immune regulator, PD-L1 interacts with its receptor PD-1 to promote tumor progression by transmitting inhibitory signals to T cells, leading to the apoptosis, suppression, anergy, and exhaustion of T cells [18]. These immunosuppressive functions occur during the induction of T cell activation in lymphoid organs, and/or during the effector phase after the migration of activated T cells to peripheral organs [19]. Since most of these functions depend on the recognition of PD-L1 on PD-1, the vast majority of PD-L1-associated studies focused on the isoforms with the IGV domain; however, little is known about the roles of Iso2, which lacks the IGV domain. Our study revealed the close relations between Iso2 and the RNA catabolism pathway, membrane protein localization, proteasome-mediated ubiquitination protein catabolism pathway, and RNA splicing, suggesting its potential roles in these biological processes. In vivo data collected from further mediation results demonstrated that Iso2 is also involved in immune regulation. However, unlike Iso1, Iso2 mainly functions by increasing T cell numbers, enhancing T cell proliferation. The increase in total T cells in the tumor microenvironment usually leads to tumor inhibition and better prognosis of tumor-burdened patients in different cancers, including liver cancer [20], which is why Iso2 improves survival. 

However, the activity of most proliferated or recruited T cells in the tumor microenvironment would be dampened by various immunosuppressive approaches hijacked or utilized by tumor cells [21]. The PD-L1 Iso1-PD1 axis is one of the most well-studied immunosuppressive pathways. No matter how much the number of T cells increases, their interaction with Iso1 triggers the transformation of exhausted T cells, destroying their capability of killing tumor cells and even promoting tumor development in some cases [21]. Our results confirmed that in all Iso2^Pos^ patients, the number of exhausted T cells was increased by the positive expression of Iso1. Therefore, in Iso1-pos tumors, merely increasing T cell numbers is not sufficient to enhance the immune killing effects, while the lack of Iso1 would guarantee the tumor-suppressive effects of increased T cells. This explains why Iso2 significantly improved the prognosis in Iso1^Neg^ patients, but this improvement was negated by the expression of Iso1.

Interestingly, besides the antagonistic effects of Iso1 on Iso2, we also observed the synergism in which the canonical immunosuppressive and tumor-promoting effects of Iso1 required positive expression of Iso2. Since the inhibition of Iso1 on T cell viability depends on its binding to PD-1 on the surface of activated T cells, the lack of activated T cells would weaken the immunosuppressive effects of Iso1 [22,23,24]. Although there were various different or even opposite functions between Iso1 and Iso2, analysis of the different associated functions of Iso1 between Iso2^pos^ and Iso2^neg^ patients indicated that immune-related pathways are key functional changes of Iso1 under the expressional switch of Iso2. Further analysis and experiments proved that Iso2 is capable of increasing T cell numbers by promoting a series of T cell-associated pathways, especially T cell proliferation. The hub gene of Iso2-positive patients includes CEBPB, an important transcription factor that regulates the expression of genes involved in immune and inflammatory response [24]. It also contains NCKAP1L, which is required for efficient T-lymphocyte migration, cell proliferation and cytokine secretion, including that of IL-2 and TNF [25]. The activated constituents would increase alongside the total T cell growth. Therefore, the presence of Iso2 determines the number of activated T cells in the microenvironment, meeting the immunosuppressive requirements for Iso1. However, in the absence of Iso2, the total number of T cells in the microenvironment is greatly reduced, and Iso1 no longer provides tumor immunosuppression. 

Actually, the process of regulating or even reversing the functions of Iso1 only requires a minimal amount of Iso2. Although Iso2 could be detected in nearly half of the patients examined, the average expressional proportion of Iso2 was much lower than that of Iso1. However, no matter how low the expression of Iso2, our in vivo data proved that even slight upregulation of Iso2 in either mouse liver or human liver cancer tissue was sufficient to increase TNF levels and trigger a severe inflammatory response, suggesting that TNF might be the key cascade factor that induces the effects of Iso2 at a very low level. It is well known that TNF induction is one of the earliest events in hepatitis, triggering a series of other cytokines that cooperate to recruit inflammatory cells, kill liver cells, and initiate a wound-healing response. TNF not only functions in liver inflammation, steatosis and fibrosis, but also plays oncogenic roles in liver cancer development, activating the proinflammatory transcription factor NF-κB to upregulate the expression of genes involved in tumor cell survival, proliferation, invasion, angiogenesis and metastasis [26]. Moreover, TNF is an important effector molecule targeting immunogenic tumor cells during tumor surveillance, which is critically required for effective priming, proliferation and recruitment of tumor-specific T cells [27]. 

In addition to TNF, our results showed that Iso2 upregulated the expression of several other members of its subfamily, such as TNF ligand superfamily member 4 (OX40L), TNF receptor superfamily member 9 (TNFRSF9) and TNF ligand superfamily member 4 (4-1BB), TNF receptor superfamily member 14 (TNFRSF14, VEM), etc (Appendix A). OX40L can be expressed by non-immune cells under inflammatory conditions, promoting the function of effector T cells and further promoting the inflammatory response [28]. Activation of 4-1BB enhanced the reactivation of T cells, and preclinical studies have shown synergistic antitumor activity between PD-L1 inhibitors and activation of the 4-1BB signaling pathway [29]. Meanwhile, we also noticed some TNF receptors were upregulated in Iso2-positive patients, such as TNFRSF18 (GITR). It has been reported that the combination of GITR and PD-1 targeting drugs enhanced anti-tumor immunity in ovarian cancer and melanoma by restoring the originally suppressed T cell activity [30]. Additionally, activation of these receptors is important for T-cell differentiation and expansion; even if some of the molecular functions are inhibited, others play a compensatory role [28]. This multi-insurance mechanism guarantees the effect of Iso2 on T cell proliferation or activation in tumors.

In recent years, with the deepening of research on the role of PD-L1 in tumor immune microenvironment and the introduction of PD-1/PD-L1 axis inhibitors, the immunotherapy of a variety of malignant tumors has undergone fundamental changes, including liver cancer. The first monoclonal antibody to be used for liver cancer was nivolumab (anti-PD-1), which demonstrated good efficacy and safety among advanced HCC patients in the CheckMate 040 test [31]. In addition, two other monoclonal antibodies against PD-L1 (Durvalumab, Atezolizumab) have also achieved excellent therapeutic outcomes. In the phase III imbrave150 study, patients treated with Atezolizumab in combination with bevacizumab (anti-VEGF) responded better than the sorafenib group, reducing the risk of OS by 56% and PFS by 40%. This combination therapy regimen was well tolerated, and the toxicity was easily controlled [32]. Recently, a Himalaya phase III study demonstrated that both durvalumab combined with tremelimumab (anti-CTLA-4) and durvalumab monotherapy significantly improved the survival of HCC patients [33]. However, several challenges have yet to be addressed, such as the low remission rate and the lack of applicable standards. Although the assessment of PD-L1 on protein levels is currently the gold standard for evaluating the treatment response, other isoforms of PD-L1 that conferred a similar antibody detection pattern in pathological analysis but opposite functions in immune response might mislead the treatment planning. Our study revealed the different roles of PD-L1 isoform and their interplay in liver cancer, providing evidence for planning a more accurate therapeutic strategy. 

According to our results, simultaneously upregulating Iso2 and downregulating Iso1 can greatly improve the prognosis of liver cancer patients. Considering the opposing prognostic effects and prognostic interplays between these two isoforms, it is necessary to validate the expression of both isoforms for PD-L1-based treatment. Additionally, we must investigate the upstream regulators and screen potential drugs that inversely regulate the expression of these two isoforms in the future, as a matter of urgency. In conclusion, our findings highlight that PD-L1 Iso2 can increase the number of T cells in the tumor microenvironment by elevating TNF levels, which is required for the tumor-suppressive effects of Iso1 in liver cancer. Our data reveal a novel interplay pattern between Iso1 and Iso2 and provide suggestions for improving the PD-L1-targeting therapeutic strategy.

## 4. Materials and Methods

### 4.1. The Data Source 

RNA sequencing data for the GTEx (https://gtexportal.org/home/, accessed on 26 January 2021) and TCGA (https://portal.gdc.cancer.gov, accessed on 26 January 2021) were retrieved from the Xena Toil RNA-Seq Recompute Compendium (https://toil.xenahubs.net, accessed on 26 January 2021), which has eliminated the batch effect between GTEx and TCGA data [34]. The normal liver samples from GTEx were defined as “normal tissues”, and the “solid tissue normal” samples in TCGA-LIHC data as “para-cancer tissues”. TCGA survival data were retrieved from the Xena TCGA data hub (https://tcga.xenahubs.net, accessed on 26 January 2021). Additionally, the validation data were derived from a Mongolian study (GSE144269) which contains the transcriptome sequencing data and matched clinical data of 68 patients with liver cancer who underwent surgery in the National Cancer Center of Mongolia between 2015 and 2016 [13]. 

### 4.2. Samples Collection

A tissue microarray consisting of tumor and para-cancer tissues from patients with liver cancer was generated as previously reported [35]. The usage of human pathological tissues and clinical data was approved by the Ethics Committee at the Shanghai Cancer Center of Fudan University (Shanghai, China) (Approval No. 050432–4-1212B). Written consent for all patients conformed to the ethical guidelines of the Helsinki Declaration. A total of 180 patients with primary HCC resected between 2010 and 2012 in the Department of Hepatic Surgery, Shanghai Cancer Center of Fudan University (Shanghai, China) were collected. None of the patients were subjected to chemotherapy, radiotherapy or other related anti-tumor therapies before surgery. Follow-up visits were conducted every three months with a median follow-up of 33.3 months (0.8–60.4 months) and were completed on 9 December 2016. 

### 4.3. Cell Culture and Transfection

Hepatocellular carcinoma cell lines Huh7 and HCC-LM3 were purchased from Shanghai Cell Bank, Chinese Academy of Sciences. Huh7 and HCC-LM3 were cultured in DMEM high glucose medium with 10% FBS in a constant temperature incubator with 5% CO_2_ at 37 °C. The PD-L1 Iso2 plasmid was constructed on the P3x-Flag vector. Transient transfections were performed by Lipofectamine 3000 reagent from Invitrogen, according to the manufacturer’s instructions. Highly purified T cells were directly isolated from the human peripheral blood using EasySep Direct Human T Cell Isolation Kit (STEMCELL Technologies) according to the manufacturer’s instructions. CD3^+^T cells were stimulated with αCD3/CD28 for 24 h. Then, cells were cocultured with transfected Huh7 cells in 48-well plates at a ratio of 5:1, or treated with the culture medium of transfected Huh7 cells. T cells were finally collected to determine the proliferation using flow cytometry after Ki-67 staining.

### 4.4. Hepatocellular Transfection via Hydrodynamic Tail-Vein Injection

The animal experiments were approved by the Ethics Committee of Fudan University (Shanghai, China) and strictly complied with the approved guidelines. The C57 mice were kept in independently ventilated mouse cages in the School of Life Sciences of Fudan University and randomly divided into groups (n = 6 in each group). Plasmids encoding Iso2 (Iso2-Flag) and controls (p3XFlag-CMV) were prepared and dissolved in sterile PBS at a work concentration of 20 μg/2 mL and then injected into the mice via the tail vein within 5–7 s using a 2 mL syringe. Five days later, the mice were sacrificed, and the livers were collected for subsequent analysis. The largest liver lobe was dissected, and the largest cross section was fixed with 4% paraformaldehyde and embedded in paraffin for subsequent HE staining, and immunohistochemical staining. 

### 4.5. Fluorescence In Situ Hybridization (FISH)

Prior to processing, slides were incubated on a 60 °C oven for 2 h. Briefly, the working solution containing the probe was applied to the pre-treated slides, which were then denatured at 37 °C for 1 h, followed by hybridization in a 40 °C humidified oven overnight. A post-hybridization wash was performed first with pre-warmed 2 × SSC (saline-sodium citrate) for 10 min at 37 °C, followed by 1 × SSC 10 min at 37 °C and 0.5 × SSC solution in a dark environment at room temperature for 10 min. The slides were stained with DAPI and mounted with coverslips. Visualization and acquisition of images of the FISH signals were performed by the automated slide scanning microscopy imaging system VS200 (Olympus). The staining intensity was graded from 0 to 3 (0, no staining; 1, weak; 2, median, 3 strong). The staining extent was graded from 0 to 4 based on the percentage of immunoreactive tumor cells (0%, 5–25%, 26–50%, 51–75%, 76–100%). The final score used in the analysis was calculated by multiplying the extent score and intensity score. 

The probe sequences targeted two mPD-L1 as follows: 

PD-L1 Iso1: CY5-ATTTGCTGAACGCCCCATACAACAAAATCA; 

PD-L1 Iso2: FITC-TTGTATGGGGCGTTCAGCAAATGCCAGTAG. 

### 4.6. Histopathology Staining

Immunohistochemistry was performed as previously described [35]. Briefly, paraffin-embedded sections were heated at 60 °C for 2 h. Slides were deparaffinized and rehydrated through xylenes and a series of graded ethanol. Heat-induced antigen retrieval was performed in retrieval solution containing EDTA using a microwave oven. The endogenous peroxidase activity was blocked with 3% H_2_O_2_ for 10 min with additional protein blocking for 15 min to minimize non-specific staining. The slides were incubated with the first antibodies overnight at 4 °C. The antigen–antibody reaction was detected using a DAKO Envision^+^ peroxidase kit (K5007) and visualized with 3,3-diaminobenzidine. Slides were lightly counterstained with hematoxylin, dehydrated in series-graded ethanol, cleared in xylene and mounted. Mouse monoclonal anti-TNF (Cell Signaling Technology, CST) were used in IHC tests as the first antibodies. Both the human liver cancer microarray sides and the mouse liver sections were stained with hematoxylin–eosin, following the manufacturer’s instructions.

### 4.7. Enzyme-Linked Immunosorbent Assay (ELISA) Analysis

ELISAs were performed to detect the levels of TNF-α secreted into supernatants from the control and Iso2-overexpression liver cancer cells. Briefly, the supernatants were collected and centrifuged at 2000× *g* for 5 min to clear the cells for further study. The levels of TNF-α were comparatively analyzed using a Human TNF alpha ELISA Kit (Abcam, Waltham, MA, USA) following the manufacturers’ instructions.

### 4.8. Isoform Recomputing

Due to the lack of isoform expression in the Mongolia liver cancer cohort (GSE144269) [13], we recomputed the RNA-seq data of this cohort to determine the isoform levels of PD-L1. The expression of the two isoforms in Mongolian HCC patients was re-analyzed according to the Hisat2-Samtools-Stringtie-Ballgown pipeline [36]. Sequencing quality was examined using FastQC. The human genome hg19 reference documents and annotations were downloaded from the Hisat2 website (http://daehwankimlab.github.io/hisat2/download/, accessed on 8 March 2022). Hisat2 was used to match the reads of the sequence to the reference genome of human hg_19, and we conducted the spliced sam file. Then, the sam file was converted into a bam file by Samtools. The mapped reads were passed to Stringtie for transcript assembly, and the final read file was applied to Ballgown. Finally, the expression of two isoforms of PD-L1 was analyzed in R. 

### 4.9. Differential Analysis

The differential profiles of mPD-L1-associated genes and pathways were analyzed using the limma package (3.20.9) between Iso2-positive and Iso2-negative patients. Adjusted *p* < 0.05 were employed as standard screening criteria. 

### 4.10. Gene Set Enrichment Analysis (GSEA) and Gene Set Variation Analysis (GSVA)

GSEA and GSVA were used to evaluate related pathways of two mPD-L1 isoforms [37]. The GO C5 gene set from the Molecular Signature Database (MSigDB) was downloaded and used for the annotation. The clusterProfiler package (4.4.4) was used for GSEA analysis and the GSVA package (1.44.2) was used for GSVA analysis. FDR q-val was used to determine whether there were significant differences.

### 4.11. Enumeration of Immune Cell Infiltration

To uncover the correlations between the expression of two PD-L1 isoforms and tumor-infiltrating immune cells, six common immune cell analysis methods including XCELL, TIMER, QUANTISEQ, MCPcounter, EPIC and CIBERSORT were utilized to evaluate the immune infiltrating situation. Each algorithm had a unique performance and advantage [38,39]. The differences in immune infiltrating cell fraction were compared between the Iso1 and Iso2 subgroups.

### 4.12. Mediation Analysis

The causal mediation analysis was performed using the Mediation package (4.5.0). To analyze the downstream factors of Iso2, Iso2 was used as the treatment, and 365 candidate genes and 77 candidate pathways were used as mediators or targets in mediation analysis. A total of 195364 (442 × 442) calculations were carried out; the mediation with significant total effects and ACMEs were screened out for further analysis. To analyze the mediations of Iso2-downstream pathways on the prognostic effects of Iso1, we fit the mediator model (linear regression) where the GSVA result of the Iso2-downstream immune pathways was modeled as a function of Iso1 levels, and the survival outcome model (survival regression) of Iso1 included the mediator and Iso1 levels.

### 4.13. Statistical Analysis

All statistical analyses were carried out in R language (4.2.0) and some results were graphically presented using Gephi (0.9.7) and Cytoscape software (3.9.1). Univariant COX analysis, Log-rank test and Kaplan–Meier curve analysis were applied for survival analysis using Survival and Survmine packages in R. Data in a bar or line graphs are presented as means ± SD or means ± SEM of at least three independent experiments. Comparisons were carried out with ANOVA or an unpaired Student’s *t*-test (* *p* <  0.05, ** *p*  <  0.01, *** *p * <  0.001).

## Figures and Tables

**Figure 1 ijms-24-06314-f001:**
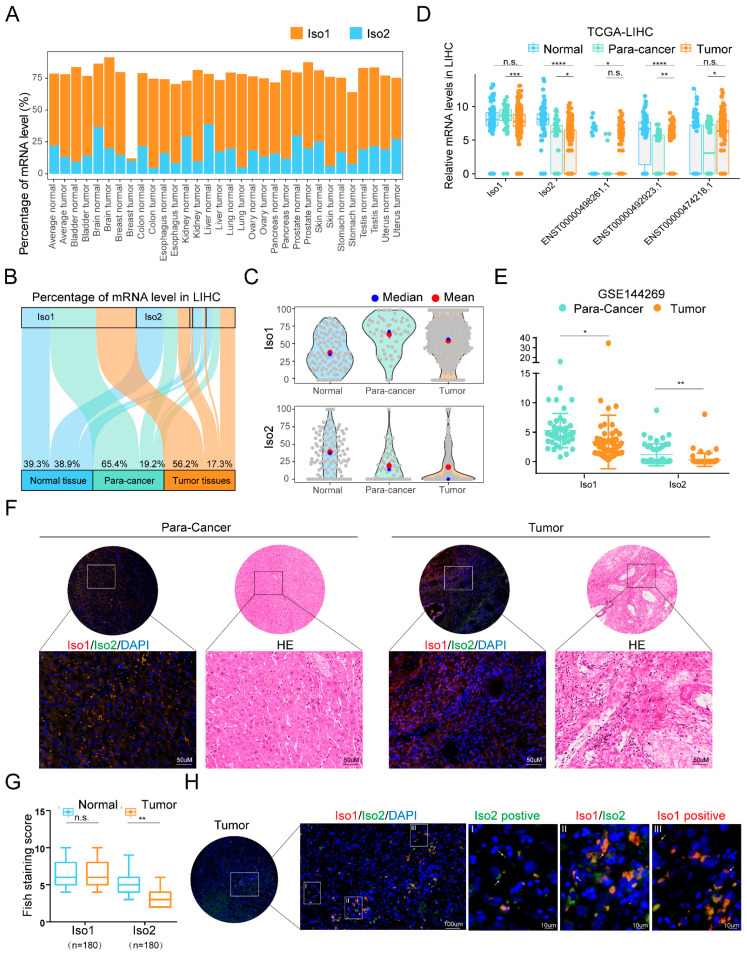
Different expression patterns of two mPD-L1 in liver cancer. (**A**) Analysis of the transcriptional isoform percentages of *PD-L1* gene in patients across 16 tissue types in TCGA and GTEx data. (**B**,**C**) Isoform percentages of *PD-L1* in normal, para-cancer and tumor tissues of liver cancer. (**D**,**E**) Relative mRNA levels of PD-L1 transcripts in the TCGA-LIHC and the Mongolian liver cancer cohorts (GSE144269). (**F**,**G**) Representative images and statistical analysis on the FISH intensity of Iso1 and Iso2 in a liver cancer tissue microarray. HE staining images were used to determine the location of Iso1 and/or Iso2-positive cells. Statistical data were presented as mean ± SD, n.s.: not significant, * *p* < 0.05, ** *p* < 0.01, *** *p* < 0.001, **** *p* < 0.0001. (**H**) Iso1 and/or Iso2-positive cells in another representative tumor tissue.

**Figure 2 ijms-24-06314-f002:**
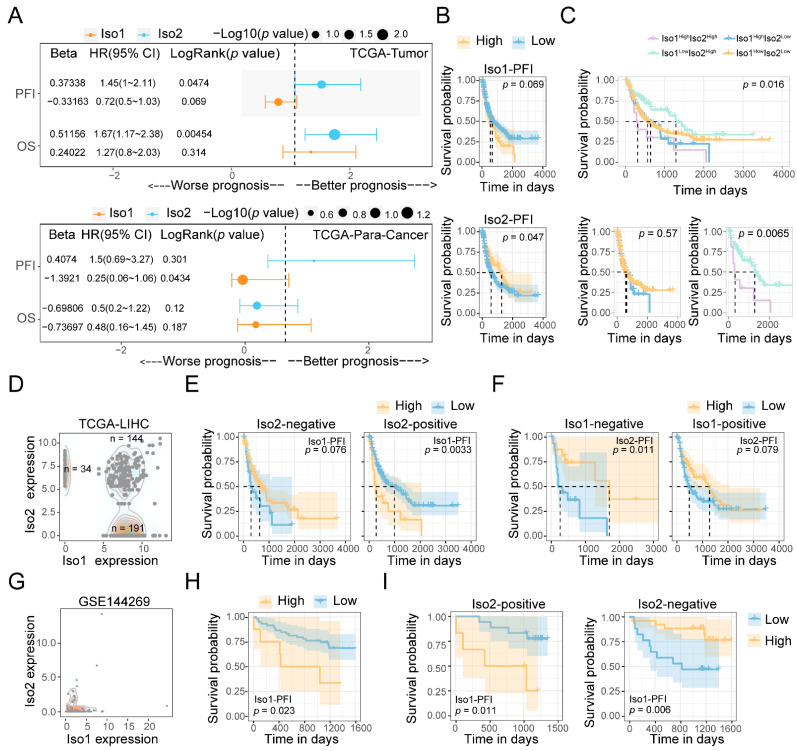
The prognostic effects of Iso2 and Iso1 and their interplays in LIHC. (**A**) Uni-Cox analysis of Iso1 and Iso2 levels on different survival indicators in both tumor and para-cancer tissues. (**B**,**C**) Kaplan–Meier analysis of the effects of Iso1 and Iso2 on PFI. The optimized cut-off points for Iso1 or Iso2 levels were calculated by Survival package. (**D**) Density plot showing the expressional distribution of Iso1 and Iso2 in TCGA-LIHC patients. (**E**,**F**) Kaplan–Meier analysis using zero as the cut-off points. (**G**) The expressional distribution of Iso1 and Iso2 in the Mongolian liver cancer cohort (GSE144269). (**H**,**I**) The roles of Iso2-positive expression in the prognostic effects of Iso1 were verified by the Mongolian data. Log-rank test was used for all Kaplan–Meier analyses.

**Figure 3 ijms-24-06314-f003:**
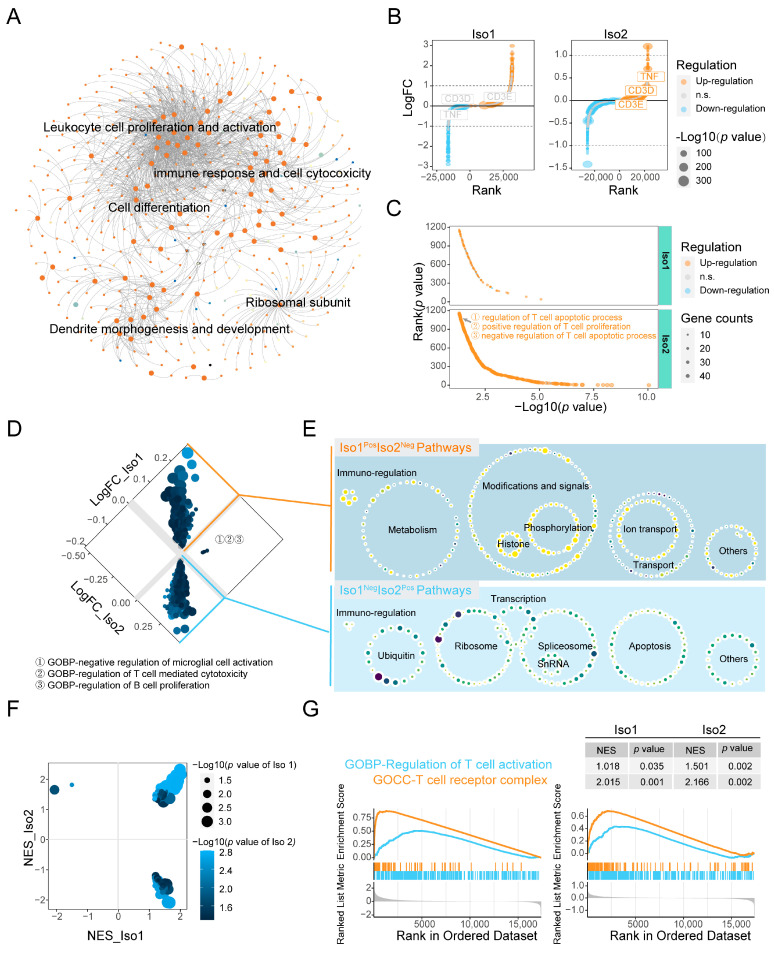
The similar and different functions between two mPD-L1 isoforms. (**A**) An integration network composed of Iso2-associated genes and pathways. (**B**) Iso1 and Iso2-associated DEGs analysis between positive and negative groups. (**C**) Functional enrichments of Iso1- and Iso2-associated DEGs. (**D**) Differential analysis of Iso1 and Iso2-associated GSVA pathways. (**E**) Distinct biology functions in Iso1^Pos^Iso2^Neg^ and Iso1^Neg^Iso2^Pos^ groups. (**F**) GSEA analysis of the associated functions of Iso1 and Iso2. (**G**) The representative GSEA results, in which Iso1 and Iso2 conferred similar effects.

**Figure 4 ijms-24-06314-f004:**
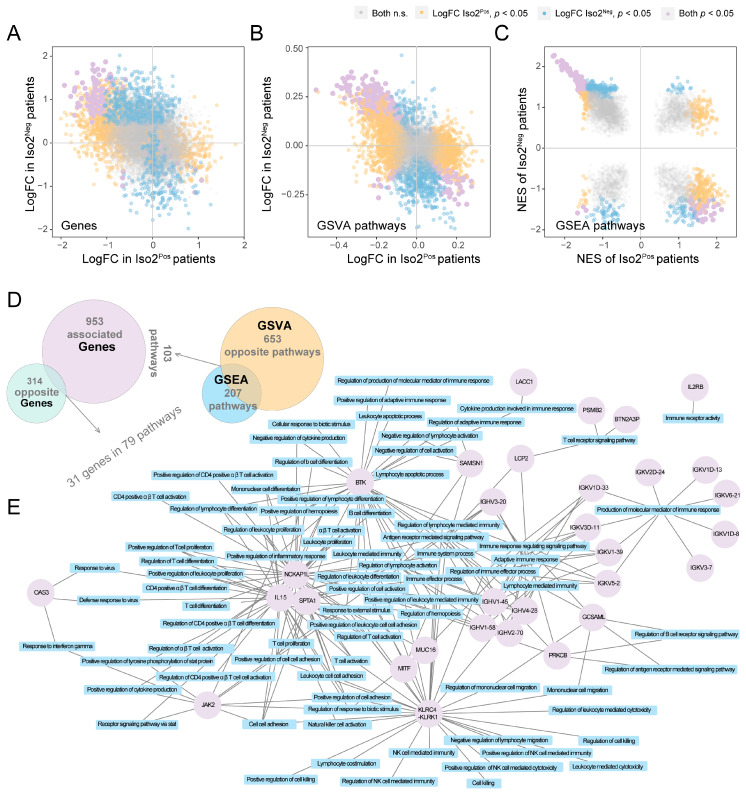
Changed or reversed functions of Iso1 under the expression switch of Iso2 from positive to negative. (**A**–**C**) Genes, GSVA and GSEA pathways significantly associated with Iso1 in Iso2^Pos^ and Iso2^Neg^ patients. (**D**,**E**) Intersected DEGs and pathways from the aforementioned analysis.

**Figure 5 ijms-24-06314-f005:**
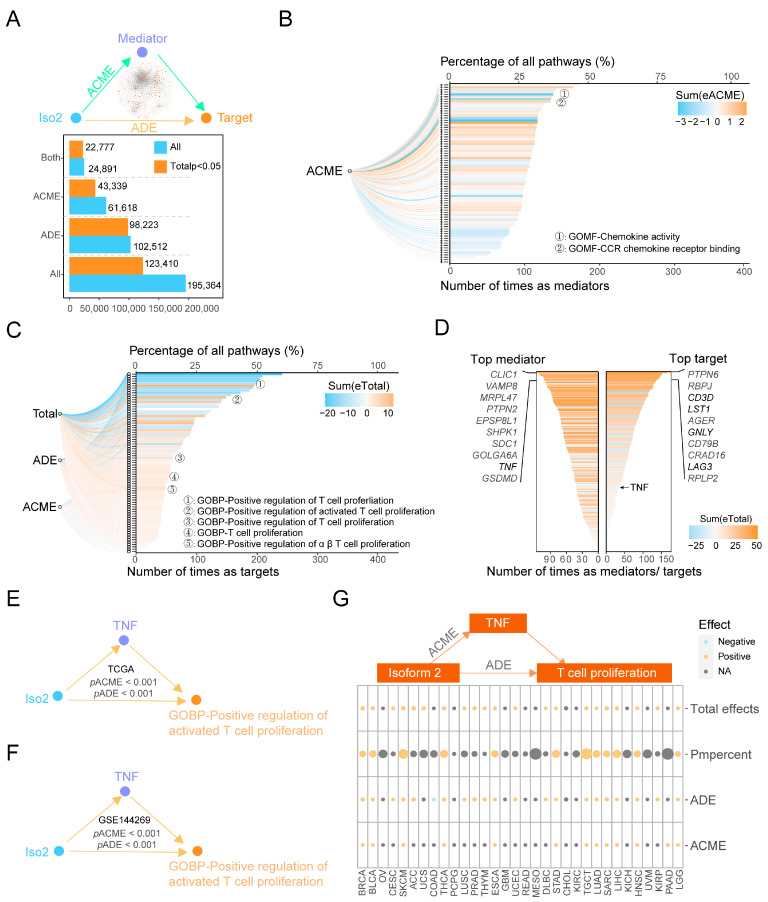
Downstream analysis of Iso2-related pathways. (**A**) Illustration of mediation analysis of Iso2 and its potential downstream factors, and a statistical overview of the mediation results. (**B**,**C**) Statistical analysis of the frequency of the downstream mediator and target pathways with significant ACMEs and total effects. (**D**) Frequency analysis of Iso2-associated genes with significant ACME ACMEs and total effects. (**E**,**F**) Representative mediation results on the regulatory axis of Iso2/*TNF*/T cell proliferation in TCGA-LIHC and the Mongolian cohorts. (**G**) The Iso2/*TNF*/T cell proliferation axis was also determined in other tumor types in TCGA data.

**Figure 6 ijms-24-06314-f006:**
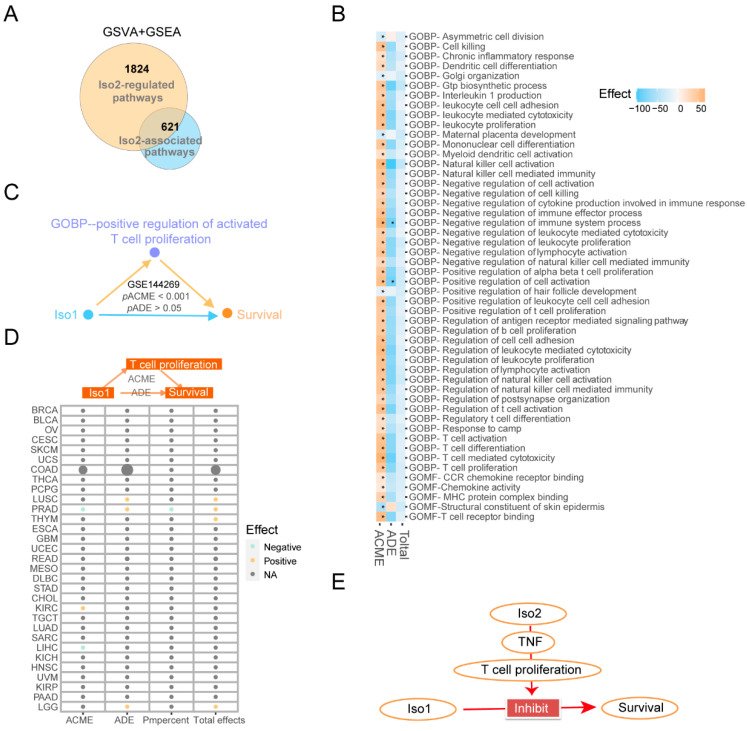
Dependency of the downstream immune pathways of Iso2 for the prognosis effect of Iso1. (**A**)Venn diagram of the intersection between the downstream immune pathways of Iso2 and the key Iso1-associated pathways changed or reversed under expressional switch of Iso2. (**B**–**D**) The mediating effects of T cell proliferation on the prognostic effects of Iso1 determined by causal mediation analysis in the Mongolian liver cancer cohort and different TCGA sub-cohorts (**E**) The prognosis promotion effect relies on the Iso2-TNF-T cell proliferation axis.

**Figure 7 ijms-24-06314-f007:**
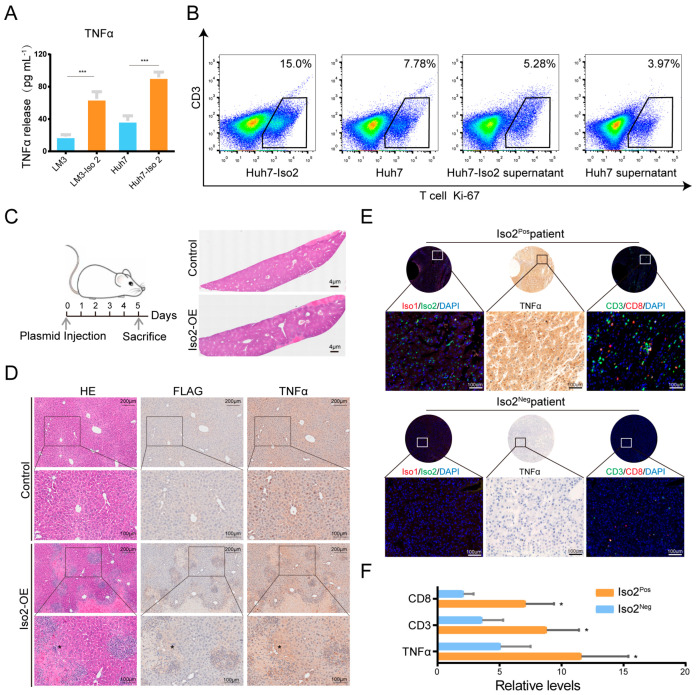
The regulation of Iso2 on TNF-α expression and T cell numbers verified by in vitro and in vivo experiments. (**A**) Increased TNF-α in protein level induced by the overexpression of Iso2. (**B**) Ki-67 staining of T cells with the indictment. (**C**,**D**) The extent of damage, inflammatory infiltration and the elevation of TNF-α triggered by slightly overexpressed Iso2 in normal liver using the mouse model of hydrodynamic tail-vein injection. The representative data were determined by HE staining and IHC staining. (**E**,**F**) The co-expression and co-localization patterns of Iso2, TNF-α and CD3^+^/CD8^+^ T cells in a liver cancer tissue microarray. Data were presented as mean ± SD, * *p* < 0.05, *** *p* < 0.001.

**Figure 8 ijms-24-06314-f008:**
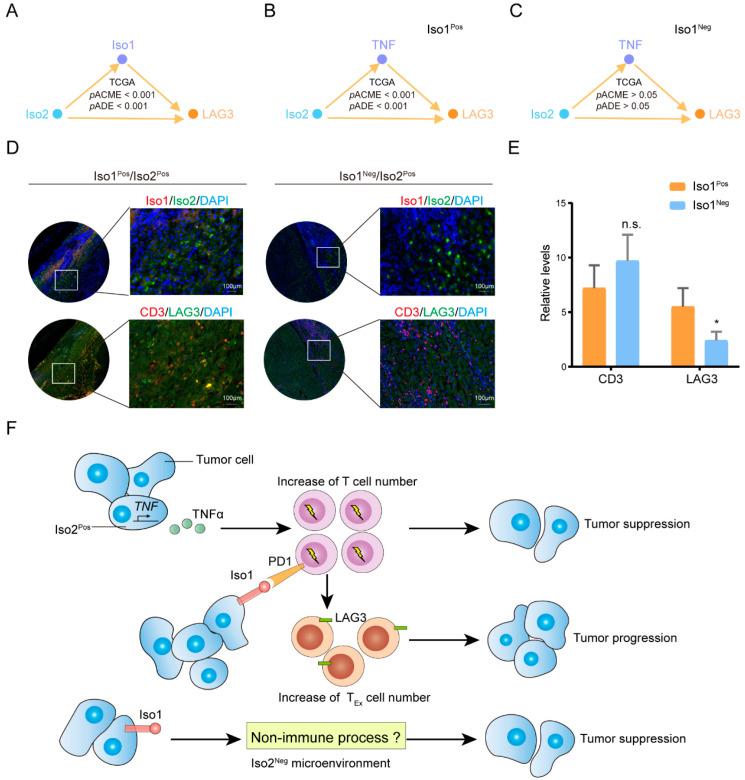
The roles of Iso1 in mediating the regulation of Iso2 on exhausted T cell. (**A**–**C**) The mediator analysis between *LAG3* and Iso2. (**D**,**E**) The immunostaining of LAG3^+^CD3^+^ T cells in Iso1^Pos^Iso2^Pos^ and Iso1^Neg^Iso2^Pos^ tumors. Data were presented as mean ± SD, n.s.: not significant, * *p* < 0.05. (**F**) Highlights of the results.

## Data Availability

Materials described in the manuscript will be freely available to any scientist wishing to use them for non-commercial purposes, without breaching participant confidentiality.

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
