# Peer review of "The Regulatory Axis of PD-L1 Isoform 2/TNF/T Cell Proliferation Is Required for the Canonical Immune-Suppressive Effects of PD-L1 Isoform 1 in Liver Cancer"

_ijms, 2023, doi:10.3390/ijms24076314_

Round 1

Reviewer 1 Report

In this report, the authors demonstrated that PD-L1 Iso2 is capable to increase the number of T cells in the tumor microenvironment via elevating TNF levels, which is required for the tumor-suppressive effects of Iso1 in liver cancer. The study contain somewhat novel concept in HCC immunology that both isoforms of PD-L1 are critical for T cell exhaustion in HCC because PD-L1 iso2 is indispensable for PD-L1 iso1 function and subsequent immune cell exhaustion. However, the methods to prove this important concept is not concrete and confirmative. Sophisticated mechanistic experiments are required at this point. However, the detailed experiments may be the next project for the authors and co-workers. Therefore, I suggest the following revision if possible for the authors. 

1) Mice model experiment may be performed with HCC orthotopic mice model to observe that tumor-infiltrating T cell responses may be accentuated or attenuated by hydrodynamic injection of the PD-L1 isoform plasmids. 

2) In vitro experiments should be performed. PD-L1 isoform plasmid transfection to hepatoma cell lines and co-cuture experiment should be strongly recommended. 

3) References are outdated and not relevant to HCC immunology field. 

Author Response

Please see the attachment, Thank you very much !

Reviewer 2 Report

Zheng et al. proposed the role of PD-L1 isoform 2 in liver cancer and focused on its role in anti-tumor T cell immune responses. Their hypothesis is intriguing and may be significant in future immunological studies and even in the clinical context, where PD-L1 is considered a biomarker for HCC immunotherapy. The usefulness and true effectiveness of PD-L1 in real-world applications still need to be determined. However, the evidence provided by the authors is not strong enough to support their conclusions, as most of the data used was from public sources rather than their own analysis.

Some specific comments are as follows:

The manuscript requires editing for formal expression in accordance with scientific journal standards.

Figure 1E) Proper citation for Fig. 1E is needed. Additionally, the statistical significance cannot be observed in Fig. 1E, contrary to the authors' statement.

Figure 1G) Representative microscopic findings for the graphs in Fig. 1G are necessary. It is important to present the difference in Iso2 expression between intratumoral and intrahepatic tissues through images.

Figure 2) Data from DSS could be moved to supplementary materials. Most HCC patients have poor liver function and cirrhosis, and it is often difficult to distinguish if the patient died from cancer or from hepatic failure from another cause. Therefore, most HCC research uses OS and PFS in clinical outcome analysis.

Figure 2) The impact of Iso2 expression on background liver tissue or paratumor tissue in the OS and PFS could also be analyzed. Not only intrahepatic immune cells, but also paratumoral or intrahepatic immune cells are involved in anti-tumor immune responses. Furthermore, most intrahepatic T cells upregulate PD-1 (which does not indicate T cell exhaustion). It may be of interest to speculate the role of Iso2 expression in clinical outcome on each site.

Figure 2) The location and expressing cell population of Iso2 could be helpful in understanding its role. This could be analyzed using flow cytometry, IF studies, or cibersort analysis if wet studies cannot be performed.

Figure 3) It would be interesting to see if the DEGs or associated pathways of Iso1 are related or share common features with PD-L1 related genes or pathways in HCC.

Figure 7) A clinical point to consider is the lack of in vivo or ex vivo validation of the hypothesis that comes from the public data analysis. For example, T cell proliferative potential (e.g., Ki-67 level), LAG3 expressing cells, and experiments for T cell function and comparison between Iso1 and Iso2 overexpressing population... are all lacking, thus the findings throughout the manuscript do not support their conclusion.

Author Response

Please see the attachment. Thank you very much !

Round 2

Reviewer 1 Report

The authors answered my questions.